# Continuous Wave-Diffuse Optical Tomography (CW-DOT) in Human Brain Mapping: A Review

**DOI:** 10.3390/s25072040

**Published:** 2025-03-25

**Authors:** Shuo Guan, Yuhang Li, Yuanyuan Gao, Yuxi Luo, Hubin Zhao, Dalin Yang, Rihui Li

**Affiliations:** 1Centre for Cognitive and Brain Sciences, Institute of Collaborative Innovation, University of Macau, Taipa, Macau SAR, China; yc27372@um.edu.mo (S.G.); yc17304@um.edu.mo (Y.L.); 2Department of Psychology, Faculty of Social Science, University of Macau, Taipa, Macau SAR, China; 3Department of Biomedical Engineering, College of Engineering, Wichita State University, Wichita, KS 67260, USA; yuanyuan.gao@wichita.edu; 4School of Biomedical Engineering, Sun Yat-sen University, Shenzhen 518107, China; luoyuxi@sysu.edu.cn; 5HUB of Intelligent Neuro-Engineering (HUBIN), CREATe, Division of Surgery and Interventional Science, University College London, London WC1H 0BW, UK; hubin.zhao@ucl.ac.uk; 6Mallinckrodt Institute of Radiology, Washington University School of Medicine, St. Louis, MO 63110, USA; yangd@wustl.edu; 7Department of Electrical and Computer Engineering, Faculty of Science and Technology, University of Macau, Taipa, Macau SAR, China

**Keywords:** continuous wave-diffuse optical tomography, brain mapping, high spatial resolution

## Abstract

Continuous wave-diffuse optical tomography (CW-DOT) has emerged as a promising non-invasive neuroimaging technique for assessing brain function. Its ability to provide brain mapping with high spatial resolution over traditional functional near-infrared spectroscopy (fNIRS) has garnered significant interest in clinical and cognitive neuroscience. In this review, we critically summarized the hardware, reconstruction algorithms, and applications of CW-DOT for human brain mapping, providing an up-to-date overview and guidelines for future studies to conduct CW-DOT studies. ScienceDirect, PubMed, Web of Science, and IEEE Xplore databases were searched from their inception up to 1 July 2024. A total of 83 articles were included in the final systematic review. The review focused on existing hardware systems, reconstruction algorithms for CW-DOT, and the applications of CW-DOT in both clinical settings and cognitive neuroscience. Finally, we highlighted current challenges and potential directions of CW-DOT in future research, including the absence of standardized protocols and a pressing need for enhanced quantitative precision. This review underscores the sophisticated capabilities of CW-DOT systems, particularly in the realm of human brain imaging. Extensive clinical and neuroscience research has attested to the technique’s anatomical precision and reliability, establishing it as a potent instrument in research and clinical practice.

## 1. Introduction

Functional near-infrared spectroscopy (fNIRS), the alternative method for functional neuroimaging in humans offers logistical advantages but at the expense of significantly inferior image quality [1,2,3]. Near-infrared light can efficiently pass through most biological tissues, allowing non-invasive imaging. When light is directed at the head, it scatters and follows a “banana-shaped” path, allowing it to reach the brain’s cortex before returning to the surface, where it can be measured [4]. The hemoglobin in the blood absorbs near-infrared wavelengths of light as it passes through the tissue, with oxyhemoglobin and deoxyhemoglobin having different absorption spectra. This enables fNIRS to track changes in blood oxygenation, which are linked to neural activity [5,6,7]. However, fNIRS has traditionally used sparse arrays of small numbers of light sources and detectors, resulting in image quality issues, such as low spatial resolution, signal localization artifacts varying with optode geometry, and contamination of cerebral signals by those from superficial tissues like the scalp and skull.

DOT addresses many of these limitations by utilizing dense arrays of light sources and detectors, significantly enhancing image quality and resolution [8,9,10,11,12,13,14,15,16,17,18,19]. Unlike traditional fNIRS systems, which may have only 3 cm source-detector pairs (Figure 1A), DOT systems can capture many measurements within a source-detector distance range of 1 to 5 cm (Figure 1B). This dense sampling allows for more accurate signal localization, improved lateral resolution, and increased signal-to-noise ratio. Additionally, the range of source-detector distances provides depth sensitivity: short-distance measurements primarily capture signals from superficial tissues to isolate brain-specific signals; while long-distance measurements convey information from deeper brain structures, to create three-dimensional maps of brain activity, offering greater specificity in targeting cortical regions.

The existence of three instruments used in optical imaging is widely known nowadays [21], including Continuous-Wave DOT (CW-DOT), Frequency-Domain (FD) DOT, and Time-Domain (TD) DOT systems. CW-DOT systems, being less complex, are more affordable to build and maintain, making them accessible for broader applications. The simplicity of CW-DOT systems also leads to more robust and reliable performance. FD and TD systems, despite their increased cost and complexity, have their own advantages. FD systems can provide additional information about the phase of the light signal. This phase information allows for more accurate quantification of the optical properties of the tissue being imaged, which is particularly useful in applications where high-precision tissue differentiation is required [22]. TD systems, on the other hand, are capable of high-resolution depth-resolved imaging. They can measure the time of flight of photons through the tissue, enabling researchers to obtain detailed information about the internal structure of the tissue at different depths [23]. Nevertheless, CW-DOT offers a practical and effective balance between complexity, cost, and imaging capability, making it a preferred choice for many functional neuroimaging applications [24,25].

The motivation for conducting a systematic assessment of CW-DOT lies in the rapid advancements within this field, contrasted by the fragmented state of the existing literature and research outcomes, which lack a cohesive summary and analysis. An urgent need exists to provide researchers with a comprehensive perspective to understand the fundamental principles, key components, and advantages of CW-DOT, thereby effectively addressing the challenges currently at hand.

Taken together, this review aimed to evaluate the current state, challenges, and future directions of CW-DOT technique by providing an in-depth understanding of its principles, components, and advantages. We also summarized the applications of CW-DOT in functional neuroimaging and other fields, demonstrating its versatility and potential. Finally, we highlighted recent developments that address existing challenges and expand the current capabilities of CW-DOT.

## 2. Methodology

### 2.1. Search Strategy

This study was conducted following the Preferred Reporting Items for Systematic Reviews and Meta-Analysis (PRISMA)-2020 guidelines [26]. The PubMed and Web of Science were searched from inception to 1 November 2024, with no language restrictions and using terms. The keywords used were (“diffuse optical tomography” OR “optical diffusion imaging” OR “DOT”) AND (“brain” OR “cerebral” OR “cortex”).

### 2.2. Prescreening and Qualifying Criteria

The prescreening criteria are based on the titles and abstracts in the database. First, duplicated articles under different titles are removed. Then, publications were excluded if they (1) were not in line with the topic, i.e., animal studies; or (2) were non-journal publications, such as reviews, conference papers, comments, dissertations, newspapers, and books. We then performed further screening by reading the full text of the articles. During this process, publications were excluded if they (1) used TD or FD devices; and (2) used only simulated data.

## 3. Results

After the pre-screening and qualification stages from the selected databases in the initial search, we obtained a total of 83 articles available for this review (Figure 2). These include 37 articles on applications, 20 on image reconstruction via various algorithms or methods, 7 on hardware validation in human brain studies, 8 on quantification or data analysis, and 11 on multimodal comparison, validation, or data fusion with different devices. From now on, DOT in this review refers to CW-DOT for simplification purposes.

Below, we provide a comprehensive review of DOT systems, from system design, data processing, and image reconstruction, to a variety of applications using DOT.

### 3.1. Structure of DOT Systems

Currently, DOT systems can be categorized into two main types: fiber-based and modular systems. The first type, “fiber-based system”, necessitates extensive cabling for data and power transmission between the headset and the control module [27]. In this architecture, the output optical data are typically digitized at a considerable distance from the detectors, which can limit system efficiency. The second type (i.e., compact) integrates the optoelectronic and control elements directly into compact head-mounted modules. This compact design enhances modularity and scalability, allowing for a more streamlined and efficient design.

#### 3.1.1. Fiber-Based Systems

In fiber-based DOT systems, the light source typically consists of dual-wavelength light-emitting diodes (LEDs) operating within the 700–850 nm spectrum, optimized for tissue penetration and controlled via time-multiplexing technology to manage light emission intervals efficiently. To detect the scattered or reflected light, sensitive detectors such as avalanche photodiodes (APDs) or silicon photodiodes are employed. APDs, in particular, require a substantial reverse bias voltage (e.g., 200 V or −150 V) to ensure optimal sensitivity.

These light sources and detectors are housed within optodes-compact, light-transmitting modules integrated into headgear to ensure stable, direct contact with the scalp. The headgear often consists of flexible materials, such as fabric caps or helmets, to maximize comfort and adaptability across different head shapes (Figure 3A,B).

Central to the system’s functionality is a control module, which includes a data acquisition board and custom-printed circuit boards (PCBs) that regulate light emission from the source and capture signals transmitted by the detectors. The control module generally communicates with a recording computer for data transmission and processing, typically via wired or wireless connections (e.g., Bluetooth). Customized DOT systems at early stages, due to the large number of sensors and the heavy weight of cables, are usually used for laboratory experiments. Notable examples of such systems include the Ultra HD-DOT system [28] and DYNOT (NIRx Medizintechnik GmbH, Berlin, Germany). Beyond customized DOT systems, traditional fNIRS devices can also achieve high-density measurement through multi-distance optode arrangements, even with a relatively small number of optodes. Examples of such devices include the NIRO-200NX (Hamamatsu Photonics, Hamamatsu, Japan), the ETG-4000 (Hitachi Medical Co., Tokyo, Japan), and the CW4-6 (TechEn Inc., Milford, MA, USA). Moreover, advancements in portable fNIRS devices have introduced wireless connectivity between the control module and the recording computer. This innovation significantly enhances wearability and portability, making them more versatile for various applications. Prominent examples of portable fNIRS devices in this category include the NIRx NIRSport2 (NIRx Medizintechnik GmbH, Berlin, Germany) and the Artinis Brite systems (Artinis Medical Systems, Elst, The Netherlands).

#### 3.1.2. Modular Systems

Modular systems consist of specialized fNIRS modules integrating key components, such as light sources—typically LEDs or Vertical Cavity Surface Emitting Lasers (VCSELs)—and detectors, including silicon photomultipliers (SiPMs) or photodiodes. Each module also incorporates driving circuits, detection electronics, and local logic controllers to manage signal processing. A base station, equipped with more advanced microcontrollers, analog-to-digital converters (ADCs), and wireless communication modules, serves as the core for data acquisition, module control, and wireless data transmission. A dedicated power supply ensures stable electrical power for the entire system, while mechanical structures are designed to fix the light sources and detectors securely, maintaining optimal contact with the scalp to ensure reliable signal acquisition.

The modular architecture offers several distinct advantages, making these systems adaptable to various research needs. The ease of expansion and customization allows researchers to combine multiple light sources and detectors into different configurations. Additionally, the lightweight and compact nature of modular components enhances user comfort and facilitates easier setup, making the system more acceptable for prolonged use, especially in clinical or developmental studies. The existing modular systems mainly include NTS and LUMO (Gowerlabs Ltd., London, UK), and Spotlight (Meta Platforms, Inc., CA, USA) (Figure 3C,D) [6,11,16].

**Figure 3 sensors-25-02040-f003:**
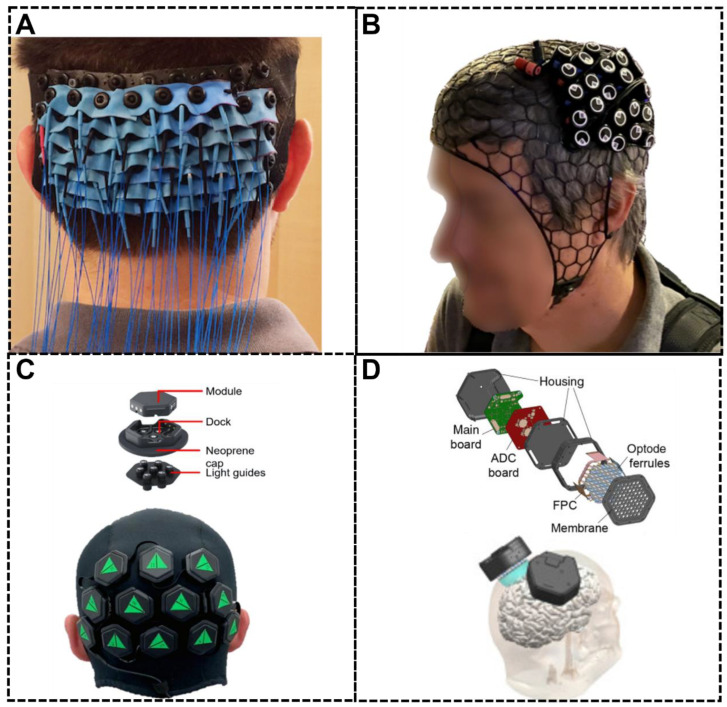
Cap design and fiber management in typical fiber-based (**A**,**B**) and modular (**C**,**D**) DOT systems. (**A**). Light weight fiber-based in-house CW-DOT system. Adapted from [29]. (**B**). NIRx NIRSport2 [6] (**C**). LUMO system: first commercially available wearable modular CW HD-DOT system. Adapted from [16]; (**D**). Spotlight: design for neuroscience and brain–computer interface (BCI) applications. Adapted from [24].

### 3.2. Imaging Reconstruction of DOT

Figure 4 shows a complete DOT image reconstruction process. The imaging reconstruction of DOT essentially consists of two parts: one is a forward modeling to calculate the light propagation and the resultant outward re-emissions at the boundary of the tissue (Figure 4D). The other is an inverse calculation searching for the distribution of optical properties (Figure 4E) [12,30,31].

#### 3.2.1. Forward Modeling

In the forward process, the goal is to model how variations in light level measurements on the surface correspond to transient changes in optical properties within the volume. This relationship can be concisely described by(1)y=Ax,
where the vectors y and x are a set of measurements and perturbed optical properties ($\mu_{a}$ and $\mu_{s}^{\prime }$) in discrete volume elements (voxels), respectively [32]. The matrix A is the Jacobian or sensitivity matrix. The vector x can be obtained from the vector y through inversion of matrix A. This sensitivity matrix is constructed from a model, termed the forward light model, derived fundamentally from the Boltzmann Transport Equation (BTE), or, equivalently in this context, the Radiative Transport Equation (RTE) [33,34,35].

Before the description of the mathematical model, it is necessary to define the energy radiance $R(r,t,\hat{s})$.(2)$$dE=R(r,t,\hat{s})\hat{s}\cdot cos\theta \cdot da\cdot d^{2}\hat{s}\cdot dt,$$
where θ is the angle between the unit direction vector $\hat{s}$ and the normal to the area element. RTE describes the energy changes radiance $R\left(r,t,\hat{s}\right)$ in time t, position r, and direction $\hat{s}$ within medium and can be expressed in a differential form:(3)$$\frac{1}{c}\frac{\partial R}{\partial t}+\hat{s}\cdot \nabla R(r,t,\hat{s})+\left(\mu_{a}+\mu_{s}\right)R(r,t,\hat{s})=\mu_{s}\int_{4\pi }f\left(\hat{s},{\hat{s}}^{\prime }\right)R\left(r,t,{\hat{s}}^{\prime }\right)d^{2}{\hat{s}}^{\prime }+Q\left(r,t,\hat{s}\right),$$
where changes in the energy $R\left(r,t,\hat{s}\right)\;$ vary by the absorption $\mu_{a}$ and scattering $\mu_{s}$, c is the speed of the light in the medium, $f(\hat{s},{\hat{s}}^{\prime })$ describes the probability of light scattering from one direction $\hat{s}$ to another direction ${\hat{s}}^{\prime }$, $Q\left(r,t,\hat{s}\right)$ is the light emission [36].

The RTE accounts for energy gains through emission and scattering redistribution and losses due to absorption. To facilitate computation, various approximation methods such as the *P*_N_ approximation, diffusion approximation (DA), [37], and the Monte Carlo method (MC) are employed. Moreover, general numerical computation methods such as the finite element modeling (FEM) and the finite difference modeling (FDM) are used to compute light propagation and predicted measurements [38]. There are packages available to create FEM, such as NIRFAST, TOAST++, or Simpleware [37,39].

#### 3.2.2. Inverse Problem

The inverse problem is to recover unknown optical properties, such as absorption and scattering coefficients, or to determine pigment concentrations and scattering characteristics from boundary data obtained through measurements. Image reconstruction can be generally formulated as an optimization problem to find $\mu_{a}$ and $\mu_{s}$ while minimizing the cost function:(4)$$\min\{{\left\|y-Ax\right\|}^{2}\}$$

The reconstruction can be achieved using the following formula:(5)$$x=A_{\lambda_{1}\lambda_{2}}^{\#}y,$$
where $\lambda_{1}$ and $\lambda_{2}$ are regularization parameters whose optimal values depend on the geometry of the source-detector grid, the noise characteristics of the imaging system, and the geometry of the anatomical model. The number of unknowns is larger than the number of measurements, regularization methods are usually used to solve inverse problems. There are generally two main approaches to solving inverse problems: linear single-step reconstruction and nonlinear iterative reconstruction.

##### Linear Approach

For a linear approximation of inverse calculation, the relationship between the optical properties and measurements of the diffusive light is linearized. The linearization of the forward process for the absolute and logarithmic values of the measured light intensities are referred to as the Born and Rytov approximations, respectively [40,41]. Then, the approach can be formulated in a manner similar to that in Equation (4) with regularization, as follows:(6)$$\underset{\delta \mu }{\min}\left\{{\left(\delta M-J\delta \mu \right)}^{T}W(\delta M-J\delta \mu )+\lambda R(\delta \mu )\right\},$$
where δM=M−F. M denotes the vector of measured light intensities, with each element $M_{i,j}$ with the ith source and jth detector pair, and F represents the vector of predicted light intensities. The regularization term R(δμ) is introduced to stabilize the ill-posed inverse problem by imposing constraints on the perturbations of the optical properties, δμ, and is weighted by the regularization parameter λ, which balances the trade-off between fitting the measured data and enforcing smoothness or prior knowledge. J is the Jacobian matrix and W is the weighting matrix, which is applied to adjust the contribution of each measurement discrepancy to the overall objective function [31].

The linearization approach is appropriate when changes in the optical properties are sufficiently small and exist in small regions. The Levenberg–Marquardt (LM) method and the perturbation method allow the construction of changes in the optical properties based on the measured light intensities on the surface [42].

This method is relatively straightforward and computationally fast; however, due to its linear assumptions, it may struggle to accurately address the complex variations in the optical properties of biological tissues, resulting in suboptimal reconstruction outcomes [40].

Recently, significant breakthroughs have been achieved based on linearization approaches in DOT. When changes in optical properties are sufficiently small and localized to small regions, the linearization approach is appropriate. Several studies have adopted spatially variant regularization and high-density source-detector arrangements to improve image quality and depth localization [43,44]. Additionally, studies have proposed a variety of methods, such as those using sparse regularization [45] and a Bayesian approach [30], to avoid errors associated with the Born and Rytov approximations. These innovations have notably enhanced the performance and reliability of linearization methods in DOT applications.

##### Nonlinear Iterative Reconstruction

Due to the inherent limitations of linear reconstruction, nonlinear reconstruction is more suitable for high-precision and high-resolution DOT applications. Linear reconstruction methods typically rely on linearized approximations of the nonlinear physical model, which can introduce significant errors under certain conditions. In contrast, nonlinear reconstruction directly handles the nonlinear light propagation model, thereby providing more accurate solutions.

The expansion of nonlinear reconstruction of Equation (4) is as follows:(7)$$\underset{\mu_{a},\mu_{s}^{\prime }}{\min}\left\{\frac{1}{2}\sum_{i=1}^{I}\sum_{j=1}^{J}\omega_{i,j}{\left(F_{i,j}\left(\mu_{a},\mu_{s}^{\prime }\right)-M_{i,j}\right)}^{2}+\gamma \cdot R\left(\mu_{a},\mu_{s}^{\prime }\right)\right\},$$

$F_{i,j}\left(\mu_{a},\mu_{s}^{\prime }\right)$ represents the operation of calculating the measured light with optical properties $\mu_{a}$ and $\mu_{s}^{\prime }$ for the given light source i and detector j to obtain the predicted measurements. And $\omega_{i,j}$ is the weight adjusting the contribution of $M_{i,j}$ to the image reconstruction. $R\left(\mu_{a},\mu_{s}^{\prime }\right)$ is a regularization term [31]. The image-reconstruction process used to solve the inverse problem is generally ill-posed [46]. Nonlinear reconstruction methods generally exhibit greater robustness and are better equipped to handle noise in measurement data and inaccuracies in prior information. Therefore, image reconstruction is performed by employing nonlinear optimization methods with iterative updating processes, such as the Newton-Raphson, quasi-Newton, and conjugate gradient methods [47,48].

### 3.3. Applications of DOT

Applications of DOT are diverse, including monitoring brain activity and hemodynamic changes in neurological and neuropsychiatric disorders, and basic neuroscience research.

#### 3.3.1. DOT in Newborn and Neonatal Research

There are nine studies that apply DOT in neonatal research, predominantly concentrating on its utility within neonatal intensive care units (NICUs), as summarized in Table 1. These investigations are primarily concerned with the monitoring of brain function in preterm infants [11,49,50]. Beyond this, several studies also explored the capacity of DOT to assess brain pathologies in infants, including hypoxic–ischemic encephalopathy [51,52]. DOT’s utility also extended to the examination of functional connectivity networks in newborns [53,54]. Two pivotal studies within this domain have shed light on infants’ reactions to emotional speech, highlighting the heightened sensitivity of the temporal–parietal cortex to positive emotional stimuli in two-month-old infants [55]. There is also a burgeoning body of research that suggests maternal anxiety during pregnancy could have a significant impact on the development of emotional processing in offspring [56].

Taken together, the collective research on DOT in neonatal studies offers a thorough perspective on the technology’s potential to monitor and assess infant brain health, as well as its significance in elucidating the intricate relationship between early-life experiences and brain development.

**Table 1 sensors-25-02040-t001:** Summary of studies of DOT for newborn and neonatal research.

First Author, Year	Age	Instrument	Channel Distance (mm)	Channels	Sampling Rate	Region of Brain	Task
Steve M. Liao, 2012 [50]	>37 weeks	HD-DOT	10, 22, 30, 36	168	10.78	occipital cortex	sleep or rest
Brian R. White, 2012 [54]	term, preterm	HD-DOT	13, 30, 40, 48	106	10.78	occipital cortex	rest
Harsimrat Singh, 2014 [52]	40 weeks	DOT-EEG	20~40	58	10	whole	rest
Silvina L. Ferradal, 2016 [49]	40 weeks	HD-DOT	10, 22, 30,36	168	10	occipital, temporal, and inferior parietal cortex	sleep or rest
Maria Chalia, 2019 [51]	40 weeks	NTS	20~40	58	10	whole	rest
Shashank Shekhar, 2019 [55]	2 months	Aalto DOT system	12~45	-	-	left temporal cortex	emotional speech
Ambika Maria, 2020 [56]	2 months	Aalto DOT system	12~45	-	-	left fronto-temporal cortex	emotional speech
Elisabetta Maria Frijia, 2021 [11]	4–7 months	LUMO	10~45	864	4.6	the superior temporal lobes and the temporopari-etal junction	video stimula
Julie Uchitel, 2023 [53]	40 weeks	LUMO	10~45	1728	10	frontal and parietal cortex	sleep

#### 3.3.2. DOT in Brain Diseases Investigation

DOT has been widely utilized to monitor brain abnormalities in various neurological and psychiatric disorders. Several studies showed that DOT can detect oxygen supply to the brain with high sensitivity and spatial resolution for patients with ischemic stroke, potentially revolutionizing clinical care during the recovery phase [57,58]. Advances in epilepsy research have demonstrated that specific hemodynamic changes precede and accompany epileptic seizures and their propagation [59]. For instance, during a finger-tapping task, diffuse areas of activation were observed in the reconstructed images of epileptic patients, while activation in healthy subjects was more focal [59]. This finding suggested that rapid functional DOT is a valuable non-invasive tool for mapping 3D cerebral blood flow dynamics. This speculation was validated by a recent study that established a portable DOT system for bedside 3D functional neuroimaging to investigate delirium in hospitalized patients. The result revealed decreased brain oxygenation and functional connectivity strength in the delirium group, even after delirium had resolved [60].

In research on psychiatric disorders, high-density DOT has captured differential brain responses in school-age children with autism spectrum disorder (ASD) compared to neurotypical individuals, revealing significant associations between brain function and dimensional measures of ASD traits [61]. DOT is not only effective in monitoring lesions but is also increasingly applied during treatment. For example, studies have used DOT to simultaneously measure changes in brain function induced by repetitive transcranial magnetic stimulation (rTMS) in both depressed and healthy subjects. Standard treatment parameters were implemented, and concurrent neuroimaging demonstrated delayed and less intense responses in patients with depression [62]. These diverse applications underscore the versatility and effectiveness of DOT in both monitoring brain lesions and assessing functional changes during various treatments and conditions.

#### 3.3.3. DOT in Visual Processing Investigation

DOT is utilized to monitor and investigate visual processing tasks in the brain. As summarized in Table 2, we reviewed 15 studies on the application of DOT in visual processing function research. Among these studies, 11 employed visual checkerboards with varying parameters as stimuli, demonstrating consistent elicit activity in the visual cortex [11,18,28,63,64,65,66,67]. These investigations further explored the effects of different checkerboard parameters on visual cortical activity, providing valuable insights into the mechanisms of visual information processing. Additionally, studies that combined DOT with functional magnetic resonance imaging (fMRI) have corroborated the spatial resolution reliability of DOT [9,10,43]. Four studies utilized videos or movies as stimuli, revealing that these more natural visual stimuli can induce more complex patterns of activity in the visual cortex [16,68,69,70]. Although videos and movies offer greater ecological validity, they also present more intricate interpretative challenges. Overall, these studies highlight the efficacy and potential of DOT technology in both monitoring and investigating the functional dynamics of the visual cortex.

**Table 2 sensors-25-02040-t002:** Summary of studies of DOT for visual processing.

First Author, Year	Age	Instrument	Channel Distance (mm)	Channels	Sampling Rate	Region of Brain	Task
Benjamin W. Zeff, 2007 [28]	23–25	HD192	13, 30, 40, 48	1200+	10	visual cortex	visual-checkboard
Joanne Markham, 2009 [63]	25–26	HD-DOT	13, 30, 40, 48	212	10.8	occipital cortex	visual-checkboard
Brian R. White, 2010 [64]	21–27	HD-DOT	13, 30, 40, 48	212	10.78	visual cortex	visual-checkboard
Brian R. White, 2010 [18]	-	HD-DOT	13, 30, 40, 48	212	10.78	visual cortex	visual stimuli
Adam T. Eggebrecht, 2012 [9]	21–30	HD-DOT	13, 30, 39, 47	1200+	10	visual cortex	visual-checkboard
Mahlega S.Hassanpour, 2014 [65]	17–30	HD-DOT	13, 30, 39, 47	1200+	10	occipital cortex	visual-checkboard
Silvina L. Ferradal, 2014 [10]	21–30	HD-DOT	13, 30, 39, 47	1200+	10	visual cortex	visual-checkboard
Adam T. Eggebrecht, 2014 [43]	21–45	HD-DOT	13, 30, 39, 47	1200+	10	occipital, temporal, motor, and frontal cortex	visual stimuli
Andrew K. Fishell, 2019 [68]	-	HD-DOT	13, 30, 39, 47	1200+	10	occipital, temporal, motor, and frontal cortex	view film
Andrew K. Fishell, 2020 [69]	8.4	HD-DOT	13, 29, 39	324	10	bilateral superior temporal gyrus	movie viewing
Kalyan Tripathy, 2021 [66]	24–54	HD-DOT	13, 30, 39, 47	1200+	10	back and side	visual stimulation
Elisabetta Maria Frijia, 2021 [11]	4–7 months	LUMO	10~45	864	4.6	the superior temporal lobes and the temporoparietal junction	video stimula
Ernesto E. Vidal-Rosas, 2021 [16]	36	LUMO	10~45	1728	5	visual cortex	visual stimulus
Jiaming Cao, 2023 [67]	-	NIRSport 2	average 27	76	5.1	occipital cortex	visual-checkboard
Kalyan Tripathy, 2024 [70]	18–81 months	HD-DOT	11, 25, 33, 39, 46	3445	-	occipital, temporal, sensorimotor cortex	movie

#### 3.3.4. DOT in Motor Function Investigation

DOT is extensively employed to investigate brain activity in response to motor functions, as detailed in Table 3. Our review of the 16 included articles revealed a predominant focused on finger-tapping tasks, which are favored for their ability to consistently elicit motor cortex activation and their suitability for standardized comparison across various studies [71,72,73,74,75,76]. Furthermore, a subset of these studies concentrated on the effects of electrical stimulation [13], finger extension [77], and squeeze tasks [12], providing a broader perspective on the motor cortex’s response characteristics under diverse movement paradigms.

Motor tasks were also applied to enhance the DOT technique in multiple aspects, such as improvements in hardware systems [24], optimization of algorithms [59,78,79,80], and experimental validation of specific tasks. These advancements have enhanced spatial resolution, depth localization, and anatomical accuracy of DOT technique in motor cortex imaging. Additionally, methods such as multimodal fusion and hierarchical Bayesian models have further bolstered the reliability and interpretability of DOT data [44,81].

**Table 3 sensors-25-02040-t003:** Summary of studies of DOT for motor function.

**First Author, Year**	**Age**	**Instrument**	**Channel Distance** **(mm)**	**Channels**	**Sampling Rate**	**Region of Brain**	**Task**
Theodore J. Huppert, 2008 [71]	-	CW4	29	-	-	primary motor cortex (M1)	finger-walking
Brian R. White, 2009 [76]	24–27	HD-DOT	13, 30, 40, 48	212	10.8	visual and motor cortex	finger tapping
Haijing Niu, 2011 [73]	18–35	DYNOT/CW5	>19/>30	65/32	-	motor	finger tapping
Anna Custo, 2010 [78]	25,34,31	CW4	>30	-	10	left hemisphere motor and pre-motor cortex	right thumb median-nerve stimulation
Jin Wook Jung, 2012 [79]	-	Oxymon MKIII	35, 78	24	10	left primary motor and somato-sensory cortex	right finger tapping go-nogo
Venkaiah C. Kavuri, 2012 [80]		CW5	30	188	100	motor cortex	finger tapping
Christina Habermehl, 2012 [13]	26.8	DYNOT	10~30	900	1.8	left motor cortex	right fingers stimulate
Meryem A. Yücel, 2012 [75]	20–60	CW4	29	-	-	primary motor area	finger tapping
Fenghua Tian, 2014 [74]	22–39	HD-DOT	16, 36	169	10.8	sensorimotor cortex	finger tapping
Okito Yamashita, 2014 [44]	-	FOIRE3000, Shimadzu	13, 29	64	5.3	left motor	right finger tapping
Okkyun Lee, 2015 [72]	-	Oxymon MKIII	35	24	10	left motor	right finger tapping go-nogo
Okito Yamashita, 2016 [81]	22–45	FOIRE 3000, Shimadzu	13, 29	64	5.3	left motor	hand movement
Danial Chitnis, 2016 [77]	23–51	NTS	8.5~85	128	2.94	motor cortex	finger extension
Xianjin Dai, 2018 [59]	-	HD-DOT	>10	-	14.4	motor	finger tapping
Daniel Anaya, 2023 [24]	33.2	spotlight	>6.5	3198 × 2	6.1	motor cortex	finger tapping
Yuanyuan Gao, 2023 [12]	-	NIRSport 2	19, 32.9	50 × 2	-	motor cortex	squeeze

#### 3.3.5. DOT in Auditory Function Investigation

Our review of five studies underscores the potential of DOT in probing auditory cortex function during both passive listening and complex speech tasks, as summarized in Table 4. Studies consistently showed that DOT can capture hemodynamic responses in the auditory cortex during passive listening to a variety of stimuli, encompassing both speech and non-speech sounds [14,69,82]. This capability to detect rapid changes in cortical activity during passive listening illuminates the temporal dynamics of auditory processing, offering insights into the early stages of sound encoding within the brain.

Furthermore, the studies that involved complex speech tasks, such as phoneme discrimination, semantic processing, and narrative comprehension, highlight DOT’s versatility in capturing higher-order auditory functions [83,84]. Specifically, by examining the spatial and temporal patterns of cortical activation across these diverse tasks, this research provides a more holistic understanding of the neural substrates that underpin speech comprehension and production.

**Table 4 sensors-25-02040-t004:** Summary of studies of DOT for auditory response.

**First Author, Year**	**Age**	**Instrument**	**Channel Distance** **(mm)**	**Channels**	**Sampling Rate**	**Region of Brain**	**Task**
Mahlega S. Hassanpour, 2015 [14]	20–32	HD-DOT	13, 30, 39, 47	1200+	10	portions of occipital, temporal, motor, and frontal cortex	speech
Mahlega S. Hassanpour, 2017 [82]	20–30	HD-DOT	13, 30, 39, 47	1200+	10	occipital, temporal, and parts of parietal, motor, and frontal cortex	auditory
Arefeh Sherafati, 2020 [83]	-	HD-DOT	13, 30, 39, 47	1200+	10	occipital, temporal, motor, and frontal cortex	hearing word
Andrew K. Fishell, 2020 [69]	8.4	HD-DOT	13, 29, 39	324	10	superior temporal gyrus	passive word listening
Mariel L. Schroeder, 2023 [84]	20–44	HD-DOT	13, 30, 39, 47	1200+	10	portions of occipital, temporal, motor, and frontal cortex	language

#### 3.3.6. DOT in Investigating Complex Cognitive Function and Brain Network

DOT’s applications in neuroscience research extend to complex task-related studies and resting-state investigations rather than function-specific research. As detailed in Table 5, four studies focused on complex tasks, while seven explored resting-state conditions.

Notably, DOT has been effectively employed to measure cortical activation in response to tactile and heat stimuli, demonstrating its potential for objective pain assessment by distinguishing between painful and non-painful stimuli [85]. This capability was further validated in another experiment that confirmed DOT’s ability to differentiate cortical responses to pain and non-painful thermal stimuli on the face [86].

In cognitive neuroscience, a study using atlas-guided DOT revealed significant age and gender differences in neural correlates of risk decision-making during the Balloon Analog Risk Task (BART) [87]. This research is particularly noteworthy, as it is the first to investigate these neural correlations in older adults, highlighting younger adults’ greater risk-taking behavior compared to older adults’ more risk-averse tendencies.

The advancement of DOT technology has also been marked by the development of a new generation of high-density, fiber-less systems. These systems have demonstrated their ability to capture brain function images during unconstrained movements, such as walking and texting, offering high sensitivity and dynamic range for 3D imaging of somatomotor cortical activation in various conditions [88]. This technological leap allows for a more comprehensive understanding of brain activity in naturalistic settings.

In the realm of functional connectivity, DOT has proven to be a valuable tool, yielding results comparable to those obtained using fMRI [89,90]. It has successfully identified the default mode network (DMN) by capturing spontaneous hemodynamic changes, showing patterns consistent with fMRI-derived networks [91,92]. Furthermore, a wearable high-density DOT system has been used to repeatedly measure resting-state functional connectivity (RSFC) in home environments, confirming stable and reliable network identification, including the visual and default mode networks [15]. Comparative studies have shown that hierarchical Bayesian algorithms for DOT outperform traditional methods in estimating RSFC when validated against fMRI signals, further solidifying DOT’s role as a robust method in neuroscience research [93,94].

**Table 5 sensors-25-02040-t005:** Summary of studies of DOT for other functions.

**First Author, Year**	**Age**	**Instrument**	**Channel Distance** **(mm)**	**Channels**	**Sampling Rate**	**Region of Brain**	**Task**
L. Becerra, 2008 [85]	18–40	CW5	30	-	-	frontal and sensory cortex	brush or heat hand
Lino Becerra, 2009 [86]	18–40	CW5	30	-	-	somatosensory cortex	heat face
Xue Wu, 2015 [90]	26	HD-DOT	10, 22, 30, 36	-	-	whole	rest
Lin Li, 2017 [87]	40 (25–40), 60 (>60)	HD-DOT	30/32.5	40/72	-	prefrontal region	BART
Estefania Hernandez-Martin, 2020 [94]	-	DYNOT232	10~40	2048	1.81	frontal cortex	rest
Takatsugu Aihara, 2020 [93]	21–38	SMARTNIRS, Shimadzu	13, 29	152	18.5	frontal and parietal area	rest
Hubin Zhao, 2020 [88]	22–45	DOT sensor module	10, 23, 28	1152	3	T7-Cz-T8	text and walk
Ali Fahim Khan, 2022 [89]	31.7	NIRxGmbH	29.8~45.6	117	6.25	whole	rest
Julie Uchitel, 2022 [15]	36	LUMO	10~45	800	-	pre-frontal and occipital regions	rest
Fan Zhang, 2023 [91]	31.7	NIRScout	30	109	6.25	whole	rest
Sruthi Srinivasan, 2024 [92]	29.1	LUMO	10~45	-	12.5,5	prefrontal, motor, and visual cortex	rest

## 4. Discussion and Future Directions

In this review, we highlighted the advancements in DOT systems and the prevalent algorithms that have significantly contributed to the field. Clinical studies have robustly demonstrated the anatomical specificity and reliability of DOT, underscoring its profound impact on clinical care. The technique’s non-invasiveness makes it particularly advantageous for use in diverse populations, including in bedside settings where traditional neuroimaging methods are impractical [51,53,95]. Additionally, DOT offers a unique advantage in capturing both superficial and relatively deep cortical activations, allowing for a more comprehensive understanding of brain function [9,18]. The flexibility in system configurations further allows for customization based on specific research and clinical needs. As evidenced by the numerous neuroscience applications, DOT stands out for its potential to fill gaps left by more established imaging technologies, offering new insights and facilitating advancements in both research and clinical domains.

Despite its potential as a non-invasive and portable alternative to fMRI, DOT faces several significant limitations and challenges that must be addressed for broader adoption in neuroscience research.

Firstly, there is a lack of standardized data acquisition configuration and data processing methods across DOT studies. That is, different research groups use varying sensor configurations, wavelengths, and pre-processing and imaging reconstruction techniques, which complicates the comparison of results and reproducibility between studies [43,77,88]. To address this, there is a clear need for the development and adoption of standardized protocols, from cap and montage design to image reconstruction [11]. By establishing universally accepted benchmarks for brain activity and functional connectivity studies using DOT, we can enhance the comparability of results and the reproducibility of findings across studies.

Secondly, despite the current DOT systems demonstrating good anatomical specificity and reliability, there remains significant room for improvement in image reconstruction algorithms. Future research can focus on developing more advanced algorithms to enhance the accuracy and efficiency of image reconstruction. For instance, leveraging deep learning and artificial intelligence techniques, such as convolutional neural networks (CNN) and generative adversarial networks (GAN), can enable the automatic learning and optimization of reconstruction parameters, thereby improving image quality and resolution [96,97]. Additionally, the development of adaptive regularization methods that dynamically adjust regularization parameters based on the characteristics of the measured data can better handle various types of noise and tissue properties [98]. Moreover, optimizing the computational efficiency of these algorithms to achieve real-time or near-real-time image reconstruction is crucial for clinical applications and dynamic monitoring [99]. These advancements will not only enhance the overall performance of DOT systems but also facilitate their broader adoption in both research and clinical settings.

Another major challenge lies in achieving a balance between the weight of the equipment, participant comfort, and the quality of the recorded signals. As DOT technology evolves towards more portable and wearable systems, the size and weight of devices becomes critical, especially for long recording sessions or studies with children and vulnerable populations [19,51]. Heavier equipment may cause discomfort and alter the participant’s natural resting state, while lighter designs might sacrifice signal quality [15,17]. Additionally, DOT systems are relatively expensive, limiting their widespread adoption in resource-constrained settings. To resolve this, future development should focus on innovative designs that maintain high-density, high-quality data acquisition without compromising participant comfort. This could involve the use of advanced materials that are lightweight yet robust, or the integration of more efficient optode arrangements that require fewer sensors while maintaining spatial coverage. Meanwhile, the development of cost-effective and high-performance DOT systems would facilitate broader application in both research and clinical settings.

The trend towards simultaneous multimodal data recording is gaining importance for a comprehensive understanding of brain function. The integration of DOT with other modalities, such as EEG, eye tracking, and physiological monitoring, has shown great potential in providing high spatiotemporal information for investigating brain activity [67]. For instance, concurrent fNIRS and EEG imaging has been instrumental in studying brain activity [100], and fNIRS with eye tracking has been used to investigate neural responses in children with genetic risk factors for ASD [101]. To capitalize on these advances, future DOT instrument development should prioritize the creation of multimodal systems that can be seamlessly integrated with a variety of sensors and modalities. We anticipate that a highly portable and multifunctional DOT system, capable of synergizing with EEG, eye tracking devices, physiology modules, accelerometers, and virtual reality devices, will open up new avenues for innovative research in the coming years.

## Figures and Tables

**Figure 1 sensors-25-02040-f001:**
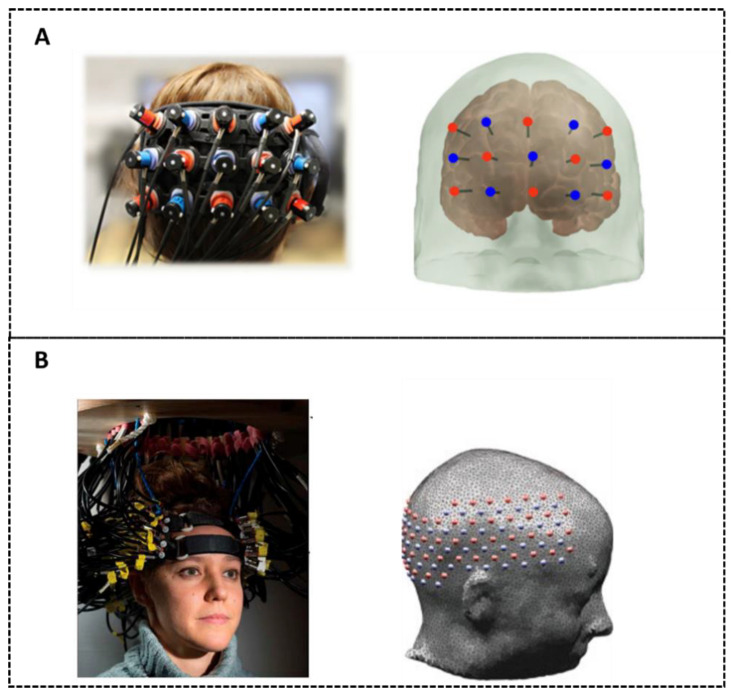
Schematic representations and optode arrangements of traditional fNIRS technique. (**A**) Adapted from [20] and the high-density DOT technique. (**B**) Adapted from [17].

**Figure 2 sensors-25-02040-f002:**
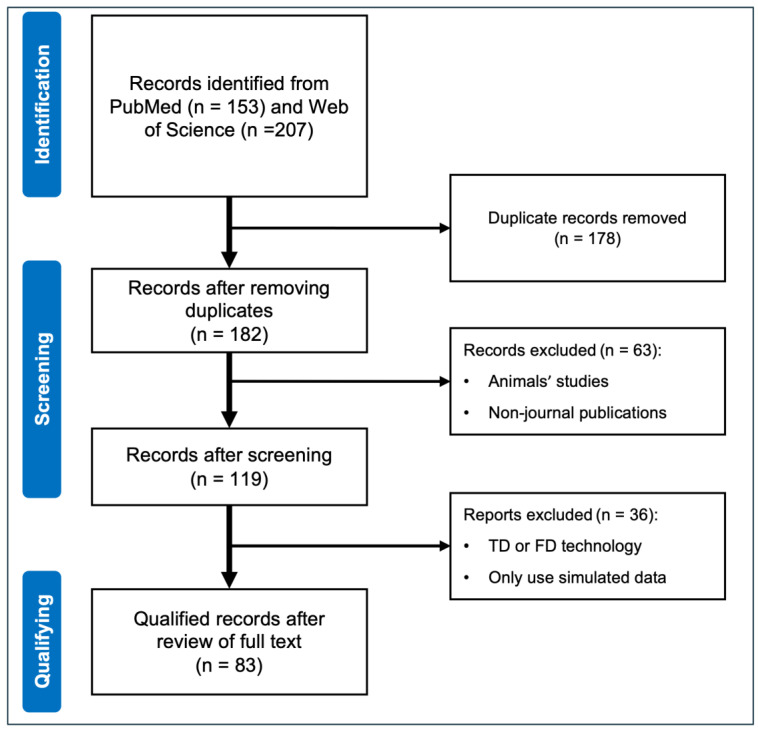
Flow chart of study selection processes.

**Figure 4 sensors-25-02040-f004:**
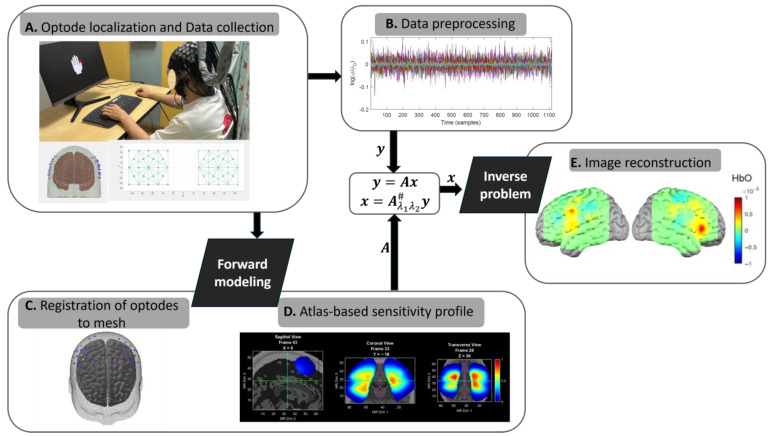
General data processing pipeline of image reconstruction in DOT. (**A**) Design of optode localization in DOT, along with diagrams of the channel layout for 3D and 2D. (**B**) During the scanning process, the raw light intensity data are preprocessed and converted into changes in optical density. (**C**) Forward modeling: a head model is created by placing the sources and detectors on the head mesh surface. (**D**) Forward modeling: a sensitivity matrix describing how the changes in light intensity at each channel location. (**E**) Inverse problem: the activated image is reconstructed by combining the properties of the sensitivity matrix (**D**) with the channel data (**B**).

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
