# Peer review of "Continuous Wave-Diffuse Optical Tomography (CW-DOT) in Human Brain Mapping: A Review"

_sensors, 2025, doi:10.3390/s25072040_

Round 1
Reviewer 1 Report
Comments and Suggestions for Authors
The authors present a review paper on the very interesting field of diffuse optical tomography applied to human brain mapping. The authors restrict themselves to continuous-wave techniques and, seeing that they still managed to select and analyze 83 articles on this specific theme, this appears to be a reasonable choice in order to keep writing compact and clear. This paper gives a comprehensive presentation of the last advances in the field, and could to my mind be published without so many corrections. I could suggest the following ones as minor suggested improvements:
- In lines 177-181, about fiber-based systems, the authors discuss the opportunity to use wireless connections to interface the control module with the computer in order “to enhance portability and ease of setup”. It could appear strange that a fiber-based system, which one can imagine as a bulky system, could open any possibility of portability. This point should be discussed a little bit more.
- In equation (7) p 9, the quantity F_i,j(mu_a,mu_a’) seems not clearly introduced (even if we guess that it is the “vector of predicted light intensities” mentioned in the previous section). I think that this point should be recalled here.
- The spatial resolution of DOT is mentioned at different places (notably lines 369 and 388) but I think that there is a lack of numbers: it could be interesting to have more quantitative information (which could be included in the different tables for instance).
Reviewer 2 Report
Comments and Suggestions for Authors
The manuscript is a thorough review of CW-DOT studies in human brain mapping. The methods are clearly stated and the qualifying 83 studies are accurately compiled.
The discussion does very well in describing the relative advantages and limitations of the technique. The highlighting of the future directions for the technique are thorough and enlightening. This raises a question as to whether adopting a standardized approach, with regard to optode placements etc, is more beneficial than a question-based approach, where optodes may be more densely packed over a structure of interest to reveal as higher detail as possible. I guess these things may not be mutually exclusive, as strong results using an optode arrangement in a brain region will lead to it being repeated, but I guess the query arises from how rapidly standardized approaches are adopted, and how much room for innovation is permitted if straying from these standardized approaches (as a new approach may answer new questions, but may risk not being attempted if the prospect of publishing is affected).
Overall, the review is strong in highlighting the uses of CW-DOT, especially in combination with other techniques.
Suggestions: In the introduction, (line 80), you highlight the advantages of CW-DOT over FD and TD versions. It may be more balanced to highlight the advantages of FD and TD, despite their increased cost and complexity, as at the moment it is not clear why anyone would choose FD or TD over CW-DOT.
